# Characterization of GEXP15 as a Potential Regulator of Protein Phosphatase 1 in *Plasmodium falciparum*

**DOI:** 10.3390/ijms241612647

**Published:** 2023-08-10

**Authors:** Hala Mansour, Alejandro Cabezas-Cruz, Véronique Peucelle, Amaury Farce, Sophie Salomé-Desnoulez, Ines Metatla, Ida Chiara Guerrera, Thomas Hollin, Jamal Khalife

**Affiliations:** 1Univ. Lille, CNRS, Inserm, CHU Lille, Institut Pasteur de Lille, U1019-UMR 9017-CIIL-Center for Infection and Immunity of Lille, 59000 Lille, France; hala.mansour@lau.edu (H.M.); veronique.peucelle@pasteur-lille.fr (V.P.); 2ANSES, INRAE, Ecole Nationale Vétérinaire d’Alfort, UMR BIPAR, Laboratoire de Santé Animale, 94700 Maisons-Alfort, France; alejandro.cabezas@vet-alfort.fr; 3Univ. Lille, Inserm, CHU Lille, U1286–Infinite–Institute for Translational Research in Inflammation, 59000 Lille, France; amaury.farce@univ-lille.fr; 4Univ. Lille, CNRS, Inserm, CHU Lille, Institut Pasteur de Lille, US 41–UAR 2014–PLBS, 59000 Lille, France; sophie.desnoulez@inserm.fr; 5Proteomics Platform Necker, Université Paris Cité–Structure Fédérative de Recherche Necker, INSERM US24/CNRS UAR3633, 75015 Paris, France; ines.metatla@inserm.fr (I.M.); chiara.guerrera@inserm.fr (I.C.G.); 6Department of Molecular, Cell and Systems Biology, University of California Riverside, Riverside, CA 92521, USA

**Keywords:** Protein Phosphatase 1, Plasmodium, malaria, GEXP15, CD2BP2, GYF domain, ribosome biogenesis

## Abstract

The Protein Phosphatase type 1 catalytic subunit (PP1c) (PF3D7_1414400) operates in combination with various regulatory proteins to specifically direct and control its phosphatase activity. However, there is little information about this phosphatase and its regulators in the human malaria parasite, *Plasmodium falciparum*. To address this knowledge gap, we conducted a comprehensive investigation into the structural and functional characteristics of a conserved *Plasmodium*-specific regulator called Gametocyte EXported Protein 15, GEXP15 (PF3D7_1031600). Through in silico analysis, we identified three significant regions of interest in GEXP15: an N-terminal region housing a PP1-interacting RVxF motif, a conserved domain whose function is unknown, and a GYF-like domain that potentially facilitates specific protein–protein interactions. To further elucidate the role of GEXP15, we conducted in vitro interaction studies that demonstrated a direct interaction between GEXP15 and PP1 via the RVxF-binding motif. This interaction was found to enhance the phosphatase activity of PP1. Additionally, utilizing a transgenic GEXP15-tagged line and live microscopy, we observed high expression of GEXP15 in late asexual stages of the parasite, with localization predominantly in the nucleus. Immunoprecipitation assays followed by mass spectrometry analyses revealed the interaction of GEXP15 with ribosomal- and RNA-binding proteins. Furthermore, through pull-down analyses of recombinant functional domains of His-tagged GEXP15, we confirmed its binding to the ribosomal complex via the GYF domain. Collectively, our study sheds light on the PfGEXP15–PP1–ribosome interaction, which plays a crucial role in protein translation. These findings suggest that PfGEXP15 could serve as a potential target for the development of malaria drugs.

## 1. Introduction

*Plasmodium falciparum* (Pf) is a unicellular parasite responsible for the deadliest form of human malaria. It poses a significant threat to global health, particularly in regions where the disease is endemic [1]. The function of Pf proteins is regulated by various post-translational modifications, with reversible protein phosphorylation being the most common protein modification observed in the parasite. Protein phosphorylation allows cells to adapt their functions rapidly in response to internal and external changes [2].

Among the Serine (Ser)/Threonine (Thr) phosphatases, Protein Phosphatase 1 (PP1c) (PF3D7_1414400) plays a crucial role in diverse cellular functions in *Plasmodium* and other organisms [3]. PP1c is a highly conserved enzyme in eukaryotes, and *Plasmodium* PP1c (PfPP1c) shares approximately 80% identity with its counterparts in mammals. Its phosphatase activity on phosphopeptides and small substrates is conserved across PP1c homologs in many species [4].

PP1c functions by associating with various regulatory partners to form holoenzymes, which specifically dephosphorylate a wide range of substrates in different cellular locations. Mammalian cells have 200 identified regulatory subunits that contribute to the specificity, location, and level of phosphatase activity of PP1c [5,6]. In Pf and in *Plasmodium berghei* (Pb), PP1c has been shown to have numerous potential regulatory partners, with hundreds of interacting proteins identified through yeast two-hybrid (Y2H) and immunoprecipitation experiments combined with mass spectrometry analysis [7,8,9].

Among the PP1c-interacting proteins, three conserved regulators (Inhibitor 2, Inhibitor 3, and LRR1) and a *Plasmodium*-specific protein Gametocyte EXported Protein 15 (GEXP15) (PF3D7_1031600) were detected as top interactors [10,11]. Biochemical studies have shown a direct interaction between PbGEXP15 and PbPP1c, increasing the phosphatase activity of PP1c in vitro [7]. Knockout of PbGEXP15 in Pb showed the vital role for the protein during the asexual life cycle and the mosquito stages, where no oocysts and sporozoites were found [7]. This phenotype could be attributed to a decrease in protein dephosphorylation due to the absence of PP1c control in the PbGEXP15 knockout line. Additionally, the crucial role of PbGEXP15 may be related to its interactome, as it was found to be associated with protein complexes involved in essential biological pathways, such as mRNA splicing and the proteasome pathway [7].

In addition to the RVxF motif located at the N-terminus of PbGEXP15, a GYF domain was identified at its C-terminal [12]. The GYF domain, characterized by the consensus sequence GP[YF]xxxx[MV]xxWxxx[GN]YF (IPR003169), is known to play a role in protein–protein interactions and is present in numerous proteins in mammals [13]. The GYF domain was initially described in the CD2 Cytoplasmic Tail-Binding Protein 2 (CD2BP2) expressed in human T cells, where it interacts with the cytoplasmic tail of the CD2 receptor, contributing to T cell activation [13]. Further studies have indicated that CD2BP2 is also present in the nucleus and may be part of the pre-spliceosomal complex [14]. Conditional gene targeting in mice revealed the essential role of CD2BP2 in embryonic development [15]. Based on reciprocal best hits (RBH) analysis, GEXP15 in *Plasmodium* is suggested to be an ortholog of human CD2BP2 [16].

Although studies on proteins functions in Pb, the most tractable of the most rodent malaria models for experimental genetics, can provide valuable insights into fundamental aspects of *Plasmodium* biology, there are limits to how much can be extrapolated to Pf [17]. For instance, targeted gene-by-gene functional studies showed that the gene encoding Schewanella-like phosphatase (shlp1) in Pf was described as likely essential for erythrocyte development by a functional screen analysis [18]. On the contrary, in Pb, shlp1 is dispensable for the development of blood stage parasites [19].

In Pf, genome-wide saturation mutagenesis suggested GEXP15 as an essential gene in the intraerythrocytic developmental cycle. However, the specific roles of this protein throughout the lifecycle of Pf are still not fully characterized. In this study, we aimed to investigate the structural and functional characteristics of GEXP15 in Pf. We employed various approaches, including comparative genomics, structural and evolutionary analyses, in vivo studies using an inducible GEXP15 knockdown line to examine cellular localization and function, and protein–protein interaction analyses to explore GEXP15′s interactors and interactome. Through these methods, we uncovered the critical interactome and potential role of GEXP15 in Pf.

## 2. Results

### 2.1. Plasmodium GEXP15 Protein Sequence Analysis

The primary structure of PfGEXP15 was compared to *Homo sapiens* (Hs) CD2BP2 (UniProt_O95400), and the alignment of their full-length protein sequences showed 23% identity (Appendix A). This low identity may be attributed to the presence of several low complexity regions (LCRs) in PfGEXP15. Pf proteins are known to have an unusually high abundance of repetitive LCRs, which often consist of amino acid repeats such as asparagine (N), lysine (K), glutamic acid (E), and aspartic acid (D). These LCRs are thought to lack any specific function [20]. Previous studies have shown that the deletion of a poly-asparagine tract in PfRPN6 did not affect protein lifetime, cellular localization, protein–protein interaction, or progression of the IDC cycle [21]. In the case of PfGEXP15, these low homology regions account for the majority of the sequence differences and the low identity between the two proteins.

Next, we compared PfGEXP15 and its potential homologs in Pb (PBANKA_0515400), *Toxoplasma gondii* (Tg) (TGGT1_217010), *Saccharomyces cerevisiae* (Sc) (known as “LIN1” in yeast, NP_012026), and Hs (UniProt_O95400). The alignment of these five proteins confirmed the presence of two conserved regions (Figure 1A). The central region of PfGEXP15 (residues 315–425) showed 25–37% identity and 39–65% similarity to the corresponding regions in PbGEXP15, HsCD2BP2, TgCD2BP2, and ScLIN1 (Figure 1B). Although the function of this central region (unknown domain, UD) is unknown, it contains conserved residues, including glycines (G315, G330 and G336) and leucines (L402, L405 and L408). The second conserved region includes the GYF domain found at the C-terminus of PbGEXP15 (residues 516–568) and PfGEXP15 (residues 713–765), as well as in Sc and Tg homologs (Figure 1C). The alignment revealed 27% identity and 52% similarity in the GYF domain between the two *Plasmodium* species. While the *Plasmodium* domain deviates from the canonical GYF consensus sequence, the substitutions involve amino acids with similar physicochemical properties (i.e., hydrophobic) to those found in the human homolog. However, two glycines, a proline (P), and a tyrosine (Y) within this domain are well conserved. The observed variations in the amino acid consensus sequence may have functional implications for the GYF-like domains.

Further analysis of PfGEXP15 identified a PP1-binding motif located in the N-terminus of the protein, similar to PbGEXP15 and human CD2BP2 (Figure 1D). This motif, KKVQF, corresponds to a canonical RVxF motif with the consensus sequence [K/R][K/R][V/I][x][F/W] [5,22] (Figure 1D). The motif is conserved in Tg but not in Sc, suggesting a loss of interaction or a different mechanism of interaction with PP1 in yeasts. Additionally, a second minimal PP1-binding RVxF motif was found in the C-terminus of PfGEXP15 (KNVYF, residues 688–692), which matches a less specific and minimal consensus sequence. The interaction of human CD2BP2 with PP1 is exclusively linked to the RVxF motif in the N-terminal end [15]. Similarly, only the first RVxF motif of PbGEXP15 was able to bind to and enhance PP1 activity, indicating a conserved PP1-binding mode between *Plasmodium* GEXP15 and CD2BP2 [7].

The identified motifs and domains were further validated using MEME Suite (Figure 2). The presence of the UD (motif 1) containing conserved glycine residues and the sequences PFN and GNY was confirmed (Figure 2). Two additional motifs were detected upstream (motif 4, residues 715–720) and downstream (motif 2, residues 729–746) of the GYF domain, but they showed high variability consistent with the degenerate consensus sequence, except for two well-conserved tryptophans and one glycine across the species (motifs 2 and 4, Figure 2). Interestingly, motif 5, composed of highly conserved amino acids, was only detected in PfGEXP15 and PbGEXP15, suggesting a potential unique function in the parasite.

### 2.2. GEXP15 3D Structure Modeling

The three-dimensional (3D) structures of PfGEXP15, PbGEXP15, and HsCD2BP2 were predicted using AlphaFold. The generated models for these three proteins exhibited well-defined structured domains along with long unfolded regions, represented as straight chains of varying lengths (Appendix A).

To enhance the accuracy of 3D predictions and address the challenges posed by unfolded regions, we conducted separated modeling for the UD and the GYF-like domains. In the UD, the presence of disordered regions hindered a complete superposition between the two *Plasmodium* proteins. However, both 3D models featured six alpha helices with a short two-stranded beta-sheet, consistent with the I-TASSER (5.1) prediction, which revealed a compact structure comprising six helices without beta sheets (Appendix A).

Regarding the GYF-like domain, both PfGEXP15 and PbGEXP15 displayed the same domain organization but with different spatial architecture (Appendix A), supporting the available NMR experimental data on the GYF-containing protein of CD2BP2 (residues 280–338) [23]. In Pb, a right angle was observed between the N-terminal helix and the beta-sheet, resulting in an almost straight orientation between the two elements. This variation may be attributed to the less structured C-terminal part of the predicted GYF domain in Pb, which may have influenced the orientation of the beta-sheet group during the optimization steps of the AlphaFold model-building process.

As for the RVxF motif, no model was generated since it often resides in unstructured regions of PP1 regulators [24]. The lack of conformation of this motif contributes to PP1 binding through a phenomenon known as “structure upon binding”. The ability of the RVxF motif to transition from an unstructured state to a structured conformation upon binding is a specific characteristic that plays a crucial role in the regulation of PP1 activity [25].

### 2.3. Distribution and Phylogenetic Analysis of CD2BP2

We investigate the distribution of CD2BP2 homologs and CD2BP2-like proteins across different Metazoan species. A total of 84 protein sequences (Appendix A) were retrieved using HsCD2BP2 as a query. Sequences showing >30% overall identity with HsCD2BP2, along with the conserved UD domain and the GYF domain, were considered CD2BP2 homologs. CD2BP2-like proteins were identified as proteins with an identity lower than 30% but possessing the UD domain and the GYF domain. CD2BP2 homologs were found in various Metazoan species, while CD2BP2-like proteins occurred in 20 phyla, including dictyostelids, fungi, choanoflagellates, rhodophytes, chlorophytes, dinoflagellates, apicomplexan parasites, and oomycetes (Figure 3). All CD2BP2 homologs had the RVxF motif and the GYF or GYF-like domains. However, CD2BP2-like proteins in Rhizaria, Plantae, and Amoebozoa lacked the RVxF motif, suggesting no PP1-binding ability. PfGEXP15 was classified as CD2BP2-like due to <30% identity but possessing the RVxF motif, UD domain, and GYF-like domain. Conservation of these domains was analyzed using MEME Suite on 84 CD2BP2 orthologs (Figure 3). Appendix A shows motifs in the UD and GYF domains. Proline, phenylalanine, glycine, asparagine (N), aspartic (D), and glutamic acid (E) residues were conserved in the UD, while motif 4 was absent in apicomplexan parasites, fungi, and oomycetes. The GYF domain has conserved N-terminal amino-acid residues (G, P, F) and C terminal residues (G, Y, F), along with other regions like WExKW (motif 7). Apicomplexa had only one of the three motifs, indicating a GYF-like domain. Using FIMO, we searched for the RVxF motif and found it in most species except platyhelminths, chlorophytes, and dinoflagellates. The motif KVTF was highly conserved in mammals, amphibians, and nematodes (Appendix A).

To assess the evolution of conserved regions between CD2BP2 and CD2BP2-like proteins, we constructed a phylogenetic tree using 66 CD2BP2 and CD2BP2-like proteins, three *Plasmodium* GEXP15 sequences, and six homologs from other Apicomplexa parasites (Appendix A). The tree revealed two major clusters. Cluster I contained vertebrate and invertebrate CD2BP2 sequences, while Cluster II included Apicomplexa, Arthropods, Mollusca, Nematodes and Cnidarians (Figure 4). Vertebrate CD2BP2 formed Cluster Ia, distinct from invertebrate CD2BP2 in Cluster Ib. Arthropod CD2BP2 sequences spanned multiple clusters, suggesting gene duplications. Cluster II consisted of Apicomplexa CD2BP2-like sequences, indicating a shared a common ancestor. Divergent tree clustering, along with shared structural features (e.g., conserved motifs and domains) and functional similarities, suggest convergent evolution between CD2BP2 and CD2BP2-like proteins.

### 2.4. Binding and Activation of PfPP1 by PfGEXP15

In a previous study, three clones corresponding to PfGEXP15 and containing the RVxF motif were identified via a Y2H screening with PfPP1c as bait [8]. In this study, a plasmid encoding a fragment of PfGEXP15 containing the first RVxF motif (8–182 residues) was used to test its ability to bind to PP1c. Only diploid cells expressing PfGEXP15 RVxF and PfPP1c were able to grow on selective media, while no yeast growth was observed with different control plasmids. This suggests a specific interaction between PfGEXP15 and PfPP1c (Figure 5A). The RVxF-dependent binding between PfGEXP15 and PfPP1c was confirmed by mutations in the binding region of PP1c (residues F255A and F256A), which prevented yeast growth under high-stringency selection.

To further confirm the direct nature of this interaction, a GST pull-down assay was performed using GST-PfPP1c and three recombinant His-tagged proteins containing the RVxF motif, UD or GYF domains produced and purified as described in Materials and Methods (Appendix A). Immunoblot analysis showed that RVxF-containing proteins bound to GST-PfPP1c but not to the GST alone (Figure 5B). An artificially dimerized form of PfGEXP15, able to bind to PfPP1c (Figure 5B), was detected and could be due to the overexpression or misfolding of the recombinant fragment produced. Neither UD-containing nor GYF-containing proteins showed binding to GST-PfPP1c, confirming the previous observation that the RVxF motif is the main contributor to the PfGEXP15–PfPP1c interaction, similar to the observation with PbGEXP15 [7].

We further examined whether the binding of PfGEXP15 affected the activity of PP1. The addition of PfGEXP15 RVxF significantly increased the dephosphorylation of p-nitrophenyl phosphate (pNPP) by PfPP1c in a concentration-dependent manner to a level similar to that of PbGEXP15 (Figure 5C). However, neither PfGEXP15 UD nor PfGEXP15 GYF showed a dose-dependent increase in PP1c activity. Only PfGEXP15 GYF at a concentration of 200 pmol/well was associated with a significant increase in PP1c activity.

Altogether, these findings demonstrate that PfGEXP15 directly and specifically interacts with PfPP1c, primarily through its RVxF motif. This binding enhances the phosphatase activity of PfPP1c in vitro.

### 2.5. Conditional Mutants, Expression, and Localization of PfGEXP15

Previous findings by Zhang et al. [18] through genome-wide saturation mutagenesis indicated that PfGEXP15 could have essential functions in the asexual stages of Pf, as no viable parasites were detected. To further investigate the role of PfGEXP15, we generated transgenic Pf parasites using an all-inclusive construction called PfGEXP15-GFP-DDD-HA (Figure 6A), based on the plasmid previously described [21]. This system enables the degradation of the tagged protein of interest in the absence of the folate analog trimethoprim (TMP). We confirmed the correct integration of the transfected plasmid at the PfGEXP15 locus by performing integration-specific PCR on the cloned population (Figure 6B), with wild-type parasites as a control (Figure 6C). The expression of PfGEXP15-GFP-DDD-HA was also detected using western blot analysis of parasite extracts probed with an anti-HA antibody, which revealed a main band at the expected size of the fusion protein (Figure 6D).

Next, we utilized the generated transgenic strain to examine the expression of PfGEXP15 throughout the asexual cycle. Western blot analysis showed that PfGEXP15 highest expression was predominantly observed during the trophozoite stage (Figure 6E). This observation aligns with RNA-seq analysis, showing a peak transcript expression during late trophozoites and early schizonts [30]. Live fluorescence microscopy analysis showed that PfGEXP15 was primarily localized in the nucleus of late trophozoite and schizont stages, with foci overlapping DNA staining (Figure 6F). This location pattern is supported by proteomic studies that detected PfGEXP15 in nuclear extracts of schizonts [31]. In the case of Pb, however, GEXP15 was also clearly detected in the parasite cytosol, suggesting species-specific functions of GEXP15 [7].

Finally, we verified the efficiency of our inducible system by western blot analysis and live fluorescence. Unexpectedly, after TMP removal, western blot examination of parasite lysates revealed that PfGEXP15 protein levels remain stable over time. Evaluation of parasite growth over several days with at least three replication cycles shows that that parasite proliferation was unaffected by the absence of TMP (Appendix A). Live fluorescence confirms that TMP had no influence on PfGEXP15 localization in the nucleus (Appendix A). The persistence of PfGEXP15, in the absence of TMP for up than four months, indicates that this method is not suitable for its degradation. These unexpected data are in line with previous studies of protein chaperones [32].

### 2.6. Identification of PfGEXP15-Interacting Proteins

To gain a better understanding of the biological roles of GEXP15 in the asexual stages of Pf, it was necessary to investigate the complexes formed by PfGEXP15. We conducted a global immunoprecipitation (IP) of PfGEXP15-HA-GFP using anti-GFP nanobodies on soluble extracts from late trophozoite and schizont stages, followed by mass spectrometry analysis (IP/MS). The parental strain was used as a control. Three biological replicates were analyzed, resulting in the identification of 1200 Pf proteins recovered from the beads (Appendix A). To refine the results, we ensured that proteins were identified in at least two out of three biological replicates, with a *p*-value (*p* < 0.05) and difference (log 2 FC) compared to the control parental strain. A total of 16 proteins were recognized, with the majority associated with the ribosomal complex (seven proteins) or RNA-binding (three proteins). STRING analysis of this interactome confirmed the enrichment of ribosome biogenesis and the translation process. However, PfPP1 did not meet the cut-off criteria in this analysis (Figure 7).

To further investigate the protein profile and determine the specific regions of GEXP15 involved in these interactions, we employed a complementary approach using pull-down experiments with recombinant proteins containing different protein domains. His-tagged proteins containing the RVxF motif, the UD, and the GYF domain were produced and coupled to nickel agarose beads, while a tetR bacterial protein served as a negative control. Soluble proteins from three independent biological replicates were incubated with the different recombinant proteins bound to beads. Prior to pull-down experiments, the presence of the tagged fragments adsorbed on the beads was confirmed by immunoblot (Appendix A). The eluted proteins were directly analyzed by MS to identify GEXP15-associated partners. A total of 312 interacting proteins were identified through the different domains (Appendix A). PF3D7_1444100, which was detected with all GEXP15 domains, and PF3D7_1206200, common between RVxF and GYF fragments, were excluded from the analysis. Principal component analysis (PCA) of the different protein domains showed distinct clusters, particularly for GYF pull-down, indicating a specific and divergent set of interactants different from RVxF and UD (Figure 8A). After filtering the proteins based on their *p*-value (*p* < 0.05) and difference (log 2 FC), nine, two, and eighty proteins were found to be significantly enriched with RVxF, UD, and GYF-containing proteins, respectively. As expected, PP1 was detected as the main interactor of the RVxF-containing protein, validating the approach (Figure 8B). Although STRING analysis of the other potential RVxF partners did not show any significant enrichment, they were associated with DNA/RNA/ATP binding and translation initiation activity.

The UD pull-down revealed only two unique proteins (Figure 8C). One of them, erythrocyte membrane-associated antigen, is present in the membrane and was excluded from analysis since GEXP15 is a nuclear protein. Therefore, exportin-7 was the only specific protein pulled down with the UD-containing protein. Exportin-7 is conserved among eukaryotes and plays a role in mediating the nuclear export of proteins into the cytoplasm. A similar function may occur in Plasmodium for the transport of GEXP15, but further investigation is needed.

For the GYF domain, 36% of the significant proteins (29/80) were found to be ribosomal subunits or ribosome-associated proteins, suggesting that this domain co-precipitated a large part of the 60S and 40S ribosomal complexes (Figure 8D). STRING analysis confirmed this observation, with enrichment of structural constituents of ribosomes (FDR = 1.73 × 10^−17^) as well as rRNA binding. Additionally, 19 biological processes were enriched, including translation (FDR = 1.33 × 10^−13^) and biosynthetic processes (Appendix A).

When comparing the results of the two approaches, three common proteins were shared between the GYF pull-down and the global PfGEXP15 IP: the 60S ribosomal proteins L26, L32, and the 40S ribosomal protein S15. However, other partners should also be considered since they share similar functions, such as small ribonucleoproteins and RNA-binding proteins. These data revealed that the most dominant network involving GEXP15 corresponds to the 40S and 60S ribosomal proteins. RVxF was found to be mainly involved in PP1 interaction, while the GYF domain played a role in the recognizing of the ribosomal machinery, unlike UD, which did not appear to be a protein-binding domain.

## 3. Discussion

In this work, we provided a better understanding of the structure and evolution of GEXP15 and its homologs in various organisms. A closer examination of these proteins highlighted three regions of particular interest. First, an RVxF motif was detected by manual inspection in the N-terminal region of PfGEXP15, PbGEXP15, TgCD2BP2, and HsCD2BP2. Using the FIMO tool, we confirmed the presence of this motif in various phyla including Apicomplexa, Metazoa, and Nematoda. This motif is known to be implicated in PP1 interaction in eukaryotes and our previous work conducted in *Plasmodium* had already established the capacity of PbGEXP15 as well as other regulators to modulate the activity of PP1 [7,10]. Here, we validated by Y2H and GST pull-down that PfGEXP15 bound to PP1, and that this interaction is RVxF-specific since the PP1 mutant and other GEXP15 regions were not able to interact. Furthermore, we demonstrated the ability of PfGEXP15 to regulate the dephosphorylation activity of PP1 through its N-terminal region containing the RVxF-binding motif. These findings confirmed the preponderant role of the RVxF motif in the interaction and regulation of PP1 by GEXP15.

Second, a conserved domain with an unknown function was identified through the in silico comparative study conducted on the different species as well as with the MEME analysis. Although our pull-down and interactome analyses showed that this domain is unlikely to be involved as a platform for protein interactions, the conservation of critical residues across distant species suggests that this UD region may play a crucial unknown role. From the MS analysis of the pull-down performed with this domain, we found only the exportin 7 (PF3D7_0910100) as a potential binder, which was detected in the nuclear fraction of Pf, suggesting its potential role in PfGEXP15 nuclear trafficking [31]. In this context, it should be noted that a previous study reported that exportin 5 is required in nuclear export of 60S ribosome subunits in human cells [33]. Further studies will be necessary for *Plasmodium* to elucidate the contribution of this domain to GEXP15 function.

Finally, our in silico study highlighted the presence of a GYF-like domain in GEXP15. The GYF domain is present in a diverse array of proteins, known to interact with proline-rich peptides, including those found in RNA-binding proteins, cytoskeletal proteins, and transcription factors [14]. Notably, the function of the GYF domain can be modulated by subtle changes in its amino acid sequence, making it a flexible region for regulating protein–protein interactions in a context-dependent manner [14]. This observation may explain why among the sequences of CB2BP2 and GEXP15 analyzed in this study, only the metazoan proteins had a GYF domain matching the currently described consensus sequence. However, despite the observed differences, the MEME analysis and 3D modeling confirmed some degree of conservation of the GYF-like domains identified in the other species, which may confer adaptation to mediate distinct protein–protein interactions.

To further investigate the functional role of GEXP15, we attempted to conditionally knock down PfGEXP15 using a degradation domain since the protein was previously suggested as essential for the development of blood-stage parasites [18]. Despite the integration of the degradation domain, confirmed by genotyping and immunoblotting, phenotypic analysis was not possible as the protein remained stable, suggesting that GEXP15 may be part of a large and stable complex. A previous study proposed that proteins not accessible to the proteasome for degradation could be a challenge for knockdown experiments [34]. Other systems can be considered, such as the Cre-LoxP system, which can be used to excise the gene of interest [35] or the *TetR*-DOZI–aptamer module repressing translation [36].

Next, we took advantage of the GFP and HA tagging of GEXP15 to follow up its localization throughout its intraerythrocytic stages. Confocal microscopy revealed that GEXP15 is highly expressed in late asexual stages, in agreement with previous transcriptomics data [12], and is localized in the parasite nucleus. In contrast to the localization of PbGEXP15 in both the nucleus and cytoplasm [7], this finding is similar to the human CD2BP2 localization [37]. This potential difference between the two *Plasmodium* species requires further investigation using electron microscopy or subcellular fractionation in order to confirm that the localization of GEXP15 is species-specific.

To gain a deeper understanding of the function of PfGEXP15, we profiled the GEXP15 interactome. A first approach based on immunoprecipitation experiments of endogenous tagged PfGEXP15-DDD-GFP-HA present in protein extracts by MS was applied to identify binding partners. This allowed the identification of 10 proteins related to one main functional group corresponding to the ribosomal complex and RNA-binding proteins.

Although PfPP1 was not detected in the PfGEXP15 IP/MS, the likelihood of this interaction via the RVxF motif was demonstrated by the use of complementary approaches such as Y2H, GST pull-down, and pull-down experiments (this study), confirming previous findings [8,9]. Supporting this is the fact that in *P. berghei* parasites, PbGEXP15 was also detected among the top PP1-interacting proteins in both schizont and gametocyte stages [9]. Further, the reciprocal IP/MS identified PbPP1 after PbGEXP15 immunoprecipitation [7]. Using this approach, the lack of PfPP1 can be due to the fact that the complex PfGEXP15-PP1 is unstable at the time point examined and/or its association is transient at this stage.

A second approach, using pull-down experiments with recombinant GYF-domain bound to beads and soluble protein extracts, revealed 29 proteins that belong to ribosomal subunits and ribosomal-associated proteins and of which three are shared with the ribosomal proteins detected in the IP-MS experiments. The limited subset of partners identified by IP and not by pull-down is not surprising as they represent different methods for interactome studies. It is known that the quantity of immunoprecipitated tagged protein, expected to be different from the quantity engaged in pull-down experiments, greatly affects MS identification. Hence, the results obtained herein can be complementary and, taken together, strongly suggest that PfGEXP15 is a ribosome-associated protein. More important is the fact that our data clearly revealed that the GYF-domain-containing protein of PfGEXP15 binds to ribosomal complex proteins, unlike the GYF domain of human CD2BP2 that has been shown to bind to spliceosomal proteins [14]. This unexpected observation suggests that the GYF-containing proteins might have diverse interactomes according to their subcellular localization, the presence and availability of species-specific partners, and/or the subtle differences in amino acids within or around the GYF domain per se. This is supported by the fact that the binding partners of GEXP15 of Pb obtained by IP experiments are different from those of Pf as they belong to spliceosomal and proteasomal core complexes which could be, at least in part, attributed to the different localization of GEXP15 in both parasites.

A closer examination of the identified proteins in the PfGEXP15 interactome showed the presence of the ribosomal RNA processing 1 homolog b (PF3D7_1414800). Interestingly, an earlier study using quantitative affinity purification followed by mass spectrometry demonstrated that human RRP1B was the most abundant partner of PP1 [38]. Moreover, it has been reported that nucleolar complexes contain both RRP1B and PP1 as components of pre-ribosomal subunit processing complexes [39]. The potential involvement of PfGEXP15 in this RRP1B-PP1 complex could therefore be envisaged. Altogether, these findings are consistent with the fact that reversible phosphorylation events via PfPP1 likely contribute to fine-tuning ribosomal biogenesis.

Despite the fact that we have not obtained direct evidence on the impact of PfGEXP15 on intraerythrocytic parasite development as the knockdown approach based on protein degradation failed, our data showing the capacity of PfGEXP15 (1) to bind and regulate PfPP1c activity, essential for *Plasmodium* survival, through its N-terminal side and (2) to interact with the ribosomal protein complex via its C-terminal side, crucial for protein translation, strongly support the essentiality of PfGEXP15. Given the functional difference between human CD2BP2 and PfGEXP15, and particularly the specific partners of the latter, identified through the GYF domain-containing protein, it would be important to determine how they interact in order to exploit specific parasite PfGEXP15–ribosome interaction for malaria drug development.

In this context, we have already shown that peptides interrupting the interaction of PP1 to its regulators via the RVxF-binding motif were able to inhibit Pf growth in vitro [40]. This proof of concept and validation of the binding of PfGEXP15 with PP1 and the ribosomal complex will open new opportunities to identify small inhibitors to disrupt this interaction network and the development of Pf.

## 4. Materials and Methods

### 4.1. Plasmid

Plasmid pGDB was a kind gift from Vasant Muralidharan. The integration plasmid, pGEXP15GDB, was synthetized by introducing a 984 bp fragment from the 3′ end of the GEXP15 ORF into pGDB between the XhoI/AvrII (New England Biolabs, Ipswich, MA, USA). PetDuet-1 was purchased from Novagen (Darmstadt, Germany).

### 4.2. Parasite Culture

The Pf3D7 strain was grown according to Trager and Jensen in RPMI 1640 medium with 10% human AB^+^ serum, in the presence of O^+^ erythrocytes [41]. Cultures were maintained at 37 °C in a humidified atmosphere (5% CO_2_, 5% O_2_, and 90% N_2_). Parasites were synchronized by successive rounds of 5% sorbitol treatment as described previously [42]. In order to isolate total proteins, parasites from infected red blood cells were purified as previously described [43].

### 4.3. MEME and FIMO Analysis

MEME Suite v5.5.1 (https://meme-suite.org/meme/tools/meme accessed on 16 March 2023) was used on the full-length sequences of GEXP15 and CD2BP2 proteins to identify conserved motifs. A maximum of 5 motifs were searched for Pf, Pb, Tg, Sc, and human sequences with maximum widths of 30 and default parameters. For the 84 CD2BP2 proteins identified, a maximum of 7 motifs were searched with the same settings. FIMO v5.5.1 (https://meme-suite.org/meme/doc/fimo.html accessed on 5 April 2023) was used to scan the RVxF motif among the 84 sequences using the consensus sequence [RK][RK][VI]X[FW] and default parameters.

### 4.4. 3D Modeling

The modeling of the PF3D7_1031600, NP_006101, and PBANKA_0515400 were carried out using Alphafold (https://alphafold.ebi.ac.uk/ accessed on 24 November 2022). I-TASSER (https://zhanggroup.org/I-TASSER/ accessed on 21 November 2022) was used additionally for the modelling of the two domains: UD (145 a.a.) and GYF (100 a.a.).

### 4.5. Phylogeny Analysis

The amino acid sequences of 66 identified CD2BP2 proteins were downloaded from the NCBI database (https://www.ncbi.nlm.nih.gov/ accessed on 12 September 2022) as well as three Plasmodium GEXP15 sequences and six homologs from Apicomplexa parasites. The species and accession numbers of each sequence is provided in Appendix A. Multiple sequence alignment of these full-length sequences was performed by Clustal Omega (https://www.ebi.ac.uk/Tools/msa/clustalo/ accessed on 11 October 2022). Then, the Neighbor-Joining method and JTT matrix-based model, implemented in MEGA X software (Version 10.2.6), were used to build a phylogenetic tree from the sequence alignment. A gamma distribution equal to one with partial deletion was used. Reliability of internal branches was assessed using the bootstrapping method (500 bootstrap replicates).

### 4.6. Yeast Two-Hybrid Assays

pGADT7-PfGEXP15 RVxF was isolated from our initial yeast-two hybrid screening [8]. Gal4-DBD-Laminin, Gal4-DBD-PfPP1c, and PfPP1c F255A/F256A were previously cloned in pGBKT7 [40]. Y2H Gold (pGADT7-PfGEXP15 RVxF) and Y187 (pGBKT7 constructs) yeast strains (Clontech, California, USA) were mated on SD-LW media. Diploids were then selected on plates lacking leucine, tryptophan and histidine (SD-LWH), and adenine (SD-LWHA) after dilutions at 1:1, 1:25, and 1:50. Plates were incubated for 4–6 days at 30 °C.

### 4.7. GST Pull-Down Assays

The coding region of the three recombinant protein fragments were PCR amplified using genomic DNA with the following primers: (1) P4–P5 for the N-terminal fragment (21–546 bp); (2) P6–P7 for the central region (625–1242 bp); and (3) P8–P9 for the C terminal portion (1878–2445 bp) (Appendix A). They were cloned into pETDuet-1 (Novagen, Darmstadt, Germany) using the In-Fusion HD Cloning system (Clontech, Mountain View, CA, USA) and transformed into One Shot^®^ BL21 Star™ (DE3) Chemically Competent *E. coli* cells (Life Technologies, Carlsbad, CA, USA) (Appendix A). The recombinant proteins were expressed in the presence of 0.5 mM IPTG at 16 °C overnight. Cells were harvested in non-denaturing buffer (20 mM Tris, 500 mM NaCl, 20 mM Imidazole, and protease inhibitor cocktail (Roche, Basel, Switzerland), pH 7.5) prior to sonication and ultracentrifugation. Then, the different pellets were resuspended for 30 min in denaturing buffer (20 mM Tris, 500 mM NaCl, 6 M Guanidine, 20 mM Imidazole, and protease inhibitor cocktail (Roche, Basel, Switzerland), pH 7.5). Recombinant proteins were purified by Ni^2+^-NTA agarose beads (Macherey Nagel, Düren, Germany) and washed with 20 mM Tris, 500 mM NaCl, and 20 mM Imidazole, pH 7.5. His-tagged proteins were eluted from beads with buffer containing 20 mM Tris, 500 mM NaCl, and 600 mM Imidazole, pH 7.5, and then the imidazole was eliminated by dialysis. The purified recombinant proteins were analyzed by western blot with anti-His antibody (1:2000 dilution) (Qiagen, Hilden, Germany) followed by HRP-labeled anti-mouse IgG (1:50,000 dilution) and quantified with a Pierce™ BCA Protein Assay Kit (Life Technologies, Carlsbad, CA, USA). GST-PfPP1c and PfPP1c were produced as previously described [7].

GST or GST-PfPP1c coupled with Glutathione-Sepharose beads (Sigma-Aldrich, Darmstadt, Germany) were saturated with 25 μg of BSA and incubated overnight at 4 °C with 2 μg of PfGEXP15 RVxF, UD, and GYF in binding buffer (20 mM Tris, 150 mM NaCl, 0.2 mM EDTA, 20 mM HEPES, 1 mM MnCl_2_, 1 mM DTT, 0.1% Triton X-100, 10% glycerol, protease inhibitor cocktail (Roche, Basel, Switzerland), and pH 7.5). After washes, proteins were analyzed by western blot, as well as 500 ng of PfGEXP15 RVxF, UD, and GYF used as inputs.

### 4.8. pNPP Phosphatase Assays

Different amounts of PfGEXP15 RVxF-, UD-, and GYF-containing proteins, described above, were preincubated with 40 pmol of PfPP1c for 30 min at 37 °C. Addition of p-nitrophenyl phosphate (pNPP) substrate (Sigma-Aldrich, Darmstadt, Germany) initiated the enzymatic reaction and after 1h of incubation, absorbance was measured at 405 nm (Thermo Scientific Multiskan FC, Marsiling Industrial Estate, Singapore). No phosphatase activity was detected with the different PfGEXP15 proteins in the absence of PP1. Two independent experiments were carried out in duplicate.

### 4.9. Transfection

To generate the PfGEXP15-HA-GFP parasite line, uninfected RBCs were transfected with 100 μg pGEXP15GDB vector then fed to wild type parasites. Drug pressure was applied 48 h after transfection, selecting for integration using 5 μM TMP (Sigma, Darmstadt, Germany) and 2.5 μg/mL Blasticidin (Calbiochem). Integration was detected after two rounds of BSD cycling after transfection. TMP was always present in the medium. Integrant clones were isolated by limiting dilution.

### 4.10. Genotype and Phenotype Analysis of Pf Transfectants

To confirm that transfected parasites contained the right integration, genomic DNA extracted (KAPA Express Extract, Kapa BioSystems, Dunedin, New Zealand) from wild or transfected parasites were analyzed by PCR using standard procedures with the primers P1–P3. Expression of the iKd PfGEXP15 protein was checked by western blotting using anti-HA (1/1000, Cell signaling C29F4, Massachusetts, USA) followed by anti-Rabbit IgG (1/20,000, Sigma, Darmstadt, Germany). Live parasites expressing PfGEXP15-GFP were analyzed by fluorescence microscopy as described below. To address the phenotype of transgenic parasites, cultures highly enriched with late trophozoites (>80%) were washed 6 times then set up ± TMP at 1% of infected red blood cells. The parasitemia were monitored up for 12 days (covering 6 life cycle) on a daily basis. After 3 and 5 cycles, viable parasites were checked for PfGEXP15 expression by live microscopy and immunoblot assays.

### 4.11. Immunoblot Assays

Parasites were suspended in Laemmli buffer and total proteins were subjected to electrophoresis in a 10% polyacrylamide gel. The proteins were transferred onto a nitrocellulose membrane (Amersham Protran 0.45 μm NC). The membrane was blocked with 5% milk (non-fat milk powder dissolved in PBS) and probed with primary antibodies (rabbit anti-HA, 1/1000 or mouse anti-His, 1/2000) diluted in the blocking buffer. The primary antibodies were followed by respective species-specific secondary antibodies conjugated to HRP (anti-rabbit, 1/20,000, Sigma) or (anti-mouse, 1/20,000, Rockland). The antibody incubations were followed by thorough washing using PBS tween 0.4%. The membranes were visualized using Dura/ Femto western blotting substrate.

### 4.12. Fluorescence Microscopy

Transgenic and parental parasites were washed then fixed with 4% paraformaldehyde and 0.0075% glutaraldehyde for 15 min at 4 °C. After PBS washing, cells were settled on Poly-L-lysine coated coverslips. The coverslips were mounted in Mowiol with DAPI (1 μg/mL) and multipoint-confocal imaging was performed with a spinning disk Live SR (stand Nikon Ti-2 combined with Live-SR module Gataca Systems, Massy, France). Figures were produced using ImageJ/Fiji software (ImageJ 1.54f, National Institutes of Health, USA).

### 4.13. Pull-Down Assays

For pull-down experiments, the RVxF-containing protein, described above, was used. For the other two recombinant proteins, shorter fragments were synthesized in order to retain the minimal functional domains based on sequence and structure analyses (UD:853–1266 bp; GYF:2053–2347 bp) (Appendix A).

The expression of His6-motifs was carried out in the *E. coli* BL21 strain in the presence of 0.5 mM IPTG at 37 °C for 2 h. Cells were harvested in lysis buffer (20 mM Tris, 150 mM NaCl, 20 mM Imidazole, Triton 1%, Lysozyme 1 mg/50 mL, DNase I, and protease inhibitor cocktail (Roche, Basel, Switzerland), pH 7.5). Recombinant proteins were purified according to manufacturer’s instructions by Ni^2+^-NTA agarose beads (QIAGEN, Hilden, Germany). Washing steps were performed with a buffer containing 20 mM Tris, 150 mM NaCl, and 20 mM imidazole, pH 7.5. Three additional washing steps with a buffer containing 20 mM Tris, 150 mM NaCl, 0.5% Triton X-100, and protease inhibitor cocktail (Roche, Basel, Switzerland), pH 7.5 were done before adding the soluble proteins to parasite extracts.

For the pull-down experiment, trophozoites/schizonts of parental wild-type parasites were suspended in 50 mM Tris, 0.5% Triton X-100, 150 mm NaCl, and protease inhibitor cocktail (Roche), pH 7.5. After ten consecutive freezing–thawing cycles and sonication, soluble fractions were obtained after repeated centrifugations at 13,000 rpm at 4 °C.

The agarose nickel beads coated with the recombinant proteins were mixed overnight at 4 °C with parasite soluble extracts in 20 mM Tris, 150 mM NaCl, 0.5% Triton X-100, and protease inhibitor cocktail (Roche, Basel, Switzerland), pH 7.5.

The beads were washed and elution was performed in Laemmli buffer. Then, after 3 min at 95 °C, samples were loaded on a 4–20% SDS-PAGE for western blot or mass spectrometry analyses. Western blots were carried out probed with anti-His mAb (1:1000, Invitrogen, Waltham, MA, USA) followed by anti-mice IgG-HRP (1:20,000, Sigma-Aldrich).

### 4.14. Sample Preparation and Immunoprecipitation

Pf-enriched trophozoite and schizont cultures of PfGEXP15-GFP-DDD-HA or parental wildtype strain (control) were used for protein extracts as described above.

Immunoprecipitation experiments were performed using 3 biological replicates of each strain. Each biological replicate contained 10 isolated pellets of trophozoites and schizonts, each purified from one culture flask of 75 cm^2^. Soluble protein extractions and immunoprecipitation assays were performed as previously described [7]. Purified parasites of each strain were suspended in 50 mM Tris, 0.5% Triton X-100, 150 mm NaCl, and protease inhibitor cocktail (Roche, Basel, Switzerland), pH 7.5. After ten consecutive freezing–thawing cycles and sonication, soluble fractions were obtained after repeated centrifugations at 13,000 rpm at 4 °C. These soluble fractions were incubated with GFP-Trap magnetic agarose (ChromoTek, Martinsried, Germany) overnight at 4 °C on a rotating wheel. The beads were washed 10 times with washing buffer containing 20 mM Tris,150 mM NaCl, 0.5% Triton X-100, and protease inhibitor cocktail (Roche, Basel, Switzerland) at pH 7.5. Elution was performed in Laemmli buffer.

### 4.15. Sample Preparation for Mass Spectrometry

S-Trap^TM^ micro spin column (Protifi, Huntington, WV, USA) digestion was performed on immunoprecipitation eluates and pull-down eluates according to the manufacturer’s instructions. Briefly, samples were supplemented with 20% SDS to a final concentration of 5%, reduced with 20 mM TCEP (Tris(2-carboxyethyl) phosphine hydrochloride), and alkylated with 50 mM CAA (chloroacetamide) for 5 min at 95 °C. Aqueous phosphoric acid was then added to a final concentration of 2.5% followed by the addition of S-Trap binding buffer (90% aqueous methanol, 100mM TEAB, pH 7.1). The mixtures were then loaded on S-Trap columns. Five washes were performed for thorough SDS elimination. Samples were digested with 2 µg of trypsin (Promega, Madison, WI, USA) at 47 °C for 2 h. After elution, peptides were vacuum dried and resuspended in 2% ACN, 0.1% formic acid in HPLC-grade water prior to MS analysis.

### 4.16. NanoLC-MS/MS Protein Identification and Quantification

The tryptic peptides were resuspended in 30 µL and an amount of 400 ng was injected on a nanoElute (Bruker Daltonics, Bremen, Germany) HPLC (high-performance liquid chromatography) system coupled to a timsTOF Pro (Bruker Daltonics, Bremen, Germany) mass spectrometer. HPLC separation (Solvent A: 0.1% formic acid in water; Solvent B: 0.1% formic acid in acetonitrile) was carried out at 250 nL/min using a packed emitter column (C18, 25 cm × 75 μm 1.6 μm) (Ion Optics, Melbourne, Australia) using a 40 min gradient elution (2 to 11% solvent B during 19 min; 11 to 16% during 7 min; 16% to 25% during 4 min; 25% to 80% for 3 min; and, finally, 80% for 7 min to wash the column). Mass spectrometric data were acquired using the parallel accumulation serial fragmentation (PASEF) acquisition method in DDA (data-dependent analysis) mode. The measurements were carried out over the m/z range from 100 to 1700 Th. The range of ion mobilities values were from 0.7 to 1.1 V s/cm^2^ (1/k0). The total cycle time was set to 1.2 s and the number of PASEF MS/MS scans was set to 6.

Data analysis was performed using MaxQuant software version 2.1.3.0 and searched with the Andromeda search engine against the TrEMBL/Swiss-Prot Pf 3D7 database downloaded from Uniprot on 10 October 2022 (5392 entries) and the *E. coli* BL21-DE3 database downloaded from Uniprot on 10 October 2022 (4173 entries). To search parent mass and fragment ions, we set a mass deviation of 10 ppm for the main search and 40 ppm, respectively. The minimum peptide length was set to 7 amino acids and strict specificity for trypsin cleavage was required, allowing up to 2 missed cleavage sites. Carbamidomethylation (Cys) was set as fixed modification, whereas oxidation (Met) and N-term acetylation (Prot N-term) were set as variable modifications. The false discovery rates (FDRs) at the peptide and protein levels were set to 1%. Scores were calculated in MaxQuant as described previously [44]. The reverse and common contaminants hits were removed from MaxQuant output as well as the protein only identified by site. Proteins were quantified according to the MaxQuant label-free algorithm using LFQ intensities, and protein quantification was obtained using at least 1 peptide per protein. Matching between runs was allowed only with IP samples.

Statistical and bioinformatic analysis, including heatmaps, profile plots, and clustering, were performed with Perseus software (version 1.6.15.0) freely available at www.perseus-framework.org accessed on 10 October 2022 [45]. For statistical comparison, we set four groups, each containing up to 3 biological replicates for the pull-down samples (Control, RVxF, UD, GYF). For the IP samples, we set two groups with 3 biological replicates each (Control, GEXP15). We then filtered the data to keep only proteins with at least 3 and 2 valid values in at least one group for pull-down and IP experiments, respectively. Next, the data were imputed to fill missing data points by creating a Gaussian distribution of random numbers with a standard deviation of 33% relative to the standard deviation of the measured values and using 3 and 1.8 SD downshift of the mean to simulate the distribution of low signal values for pull-down and IP datasets, respectively. We then performed an ANOVA test (FDR < 0.05, S0 = 1) for the pull-down samples and a statistical *t*-test (FDR < 0.05, S0 = 0.1) for IP samples. Hierarchical clustering of proteins that survived the test was performed in Perseus on LFQ intensities expressed on a logarithmic scale after z-score normalization of the data using Euclidean distances.

## Figures and Tables

**Figure 1 ijms-24-12647-f001:**
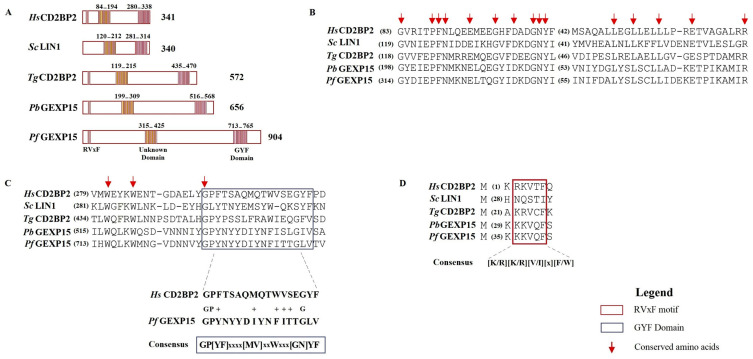
In silico analysis of *Plasmodium* GEXP15 and CD2BP2 homologs. (**A**) GEXP15 amino acid sequences from Pb and Pf were aligned with CD2BP2 from *H. sapiens*, *S. cerevisiae* and *T. gondii* using the MAFFT alignment program. A schematic representation of relevant motifs alongside their positions. (**B**) Multiple protein sequence alignment of an unknown conserved domain (UD). (**C**) GYF and GYF-like domain alignment with the consensus sequence. (**D**) Multiple alignments of the conserved RVxF motif, represented above its consensus sequence. Arrows show the conserved amino acid residues.

**Figure 2 ijms-24-12647-f002:**
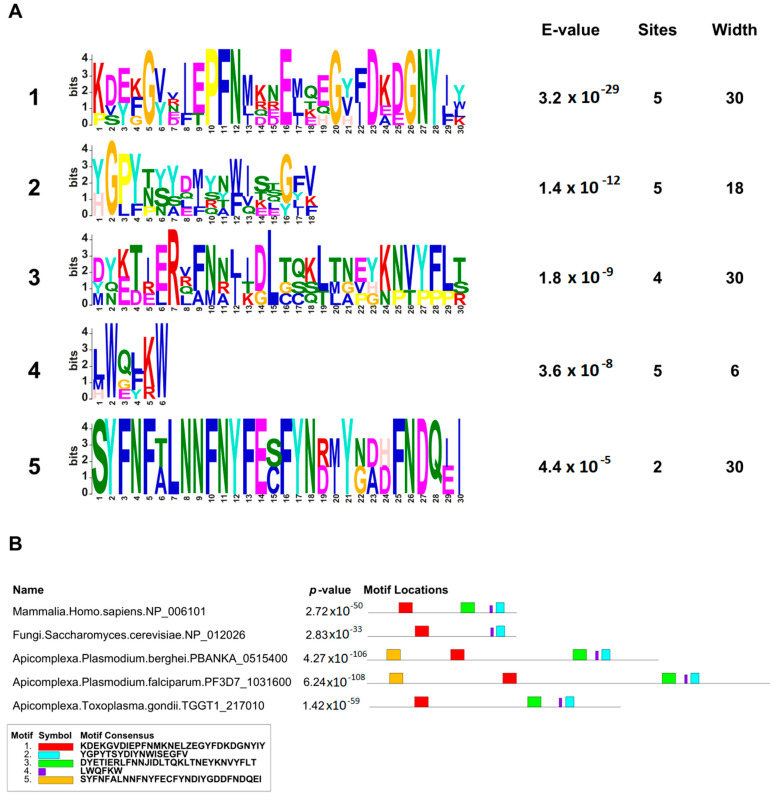
MEME motif search of GEXP15 and CD2BP2 proteins. (**A**) The 5 most significant sequence logos identified by MEME are represented, as well as their respective E-values, number of sites and widths. The height and size of the letters represent the amino acid frequency. (**B**) Distribution of these motifs across HsCD2BP2, ScLIN1, PbGEXP15, PfGEXP15, and TgCD2BP2. The color of each motif is indicated in part A. *p*-value and consensus sequence are also reported.

**Figure 3 ijms-24-12647-f003:**
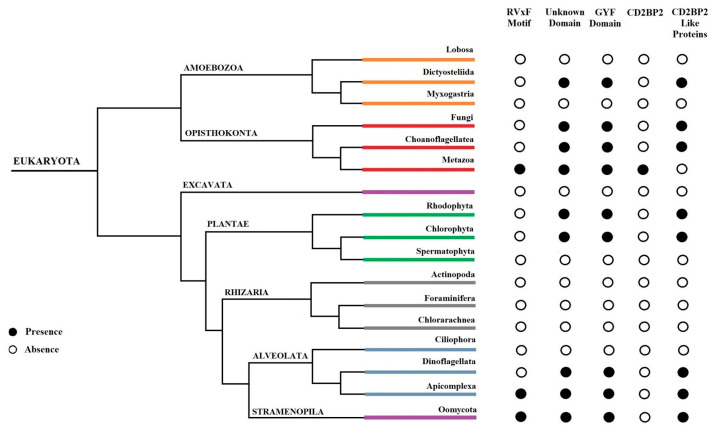
Distribution of CD2BP2 homologs and their domains in eukaryotes. The figure displays the distribution of CD2BP2 in the phylogenetic tree of life. Open and closed circles represent absence and presence of the RVxF motif, unknown domain, GYF domain, CD2BP2 homologs, and CD2BP2-like proteins, respectively. For fungi, *Saccharomyces* species were the main ones considered.

**Figure 4 ijms-24-12647-f004:**
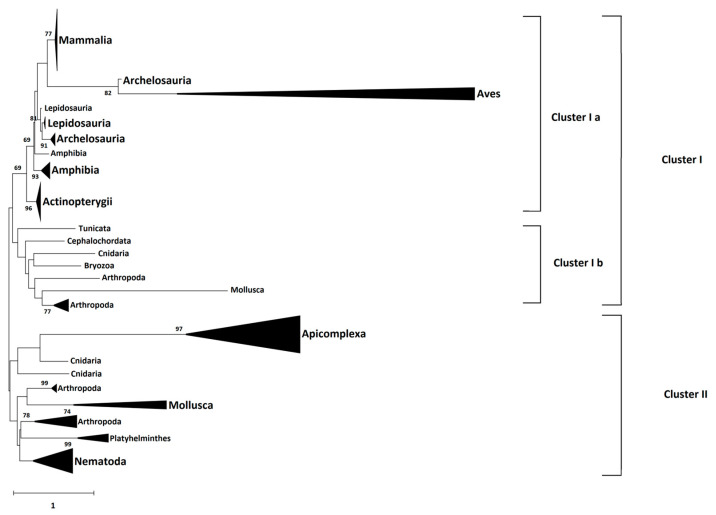
Phylogenetic tree of CD2BP2 and GEXP15 proteins. The evolutionary history was inferred using the Neighbor-Joining method [26]. The optimal tree is shown. The percentage of replicate trees in which the associated taxa clustered together in the bootstrap test (500 replicates) are shown next to the branches [27]. The tree is drawn to scale, with branch lengths in the same units as those of the evolutionary distances used to infer the phylogenetic tree. The evolutionary distances were computed using the JTT matrix-based method [28] and are in the units of the number of amino acid substitutions per site. The rate variation among sites was modeled with a gamma distribution (shape parameter = 1). This analysis involved 66 amino acid sequences. All positions with less than 95% site coverage were eliminated, i.e., fewer than 5% alignment gaps, missing data, and ambiguous bases were allowed at any position (partial deletion option). There was a total of 213 positions in the final dataset. Evolutionary analyses were conducted in MEGA 11 [29].

**Figure 5 ijms-24-12647-f005:**
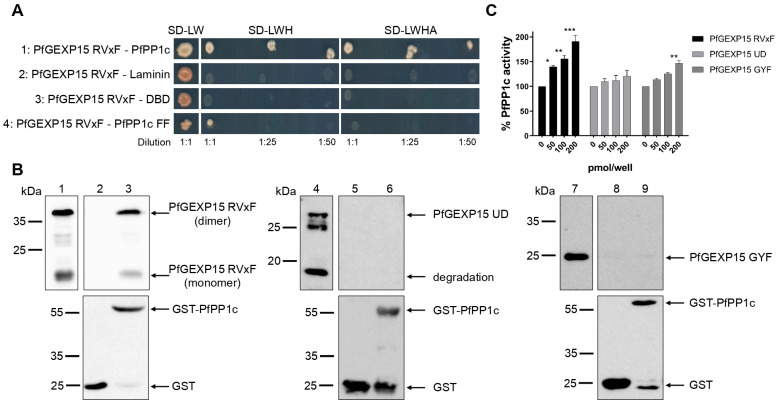
Interaction of PfGEXP15 with PfPP1c and its regulatory effect on the phosphatase activity. (**A**) Yeast two-hybrid assay. pGADT7-PfGEXP15 RVxF was mated with pGBKT7-PfPP1c (lane 1), pGBKT7-Laminin (lane 2), pGBKT7-DBD (lane 3), and pGBKT7-PfPP1c F255A F256A (FF) (lane 4). Yeast diploids were plated on SD-LW, SD-LWH, and SD-LWHA selective media and interactions were identified by growth of undiluted and diluted (1:25 and 1:50) cultures. (**B**) GST pull-down assay. Lane 1 shows the input of 6-His PfGEXP15 RVxF (500 ng) and, in lanes 2 and 3, the eluted proteins (2 μg) after incubation with GST alone or GST-PfPP1c, respectively. The recombinant proteins 6-His PfGEXP15 UD and 6-His PfGEXP15 GYF are loaded in the same conditions in lanes 4–5–6 and 7–8–9, respectively. Immunoblots are revealed with mAb anti-His (upper panel) and anti-GST (lower panel). (**C**) pNPP-phosphatase assay. The recombinant proteins 6-His PfGEXP15 RVxF, 6-His PfGEXP15 UD, and 6-His PfGEXP15 GYF were incubated at different concentrations with PfPP1c for 30 min at 37 °C before the addition of para-nitrophenyl phosphate (pNPP). The linear formation of the dephosphorylated product, para-nitrophenol, was measured by optical density after 1h at 37 °C. Results are reported as mean ± SD of the percent relative activity (*n* = 2 in duplicate). Significance was determined by Kruskal–Wallis with Dunn’s post-hoc test: * *p* < 0.05, ** *p* < 0.01, *** *p* < 0.001. The detection of free GST in lanes 3, 6, and 9 could be attributed to non-specific cleavage or protease activity.

**Figure 6 ijms-24-12647-f006:**
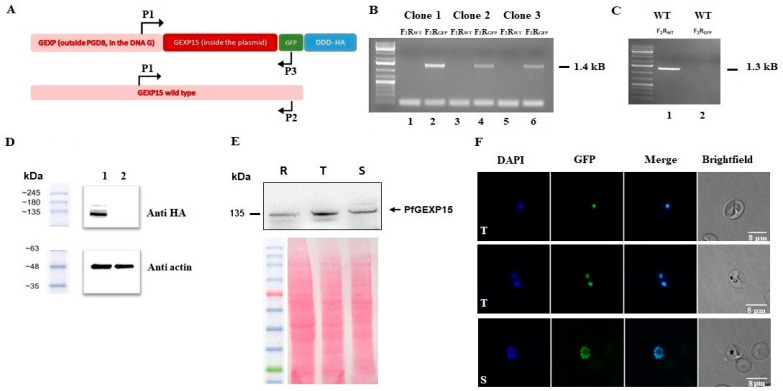
Expression and localization of PfGEXP15-GFP-DDD-HA. (**A**) Schematic representation of the pGDB construct and the primers used to check plasmid integration. The GEXP15 is tagged with DDD, GFP, and HA tags. (**B**) Diagnostic PCR analysis of tagged GEXP15 clones. Lanes 1–6 correspond to gDNA extracted from transfected parasites. Lanes 1, 3, and 5 represent the detection of the wild-type (WT) locus; lanes 2, 4, and 6 correspond to the integration of the construct. (**C**) Diagnostic PCR analysis of WT parasites. Lane 1 represents the detection of the WT locus; lane 2 corresponds to the integration of the construct. (**D**) Western blot analysis representing the soluble protein extract from transgenic PfGEXP15 in lane 1 and WT parasites as negative control in lane 2. They were revealed with mAb anti-HA rabbit. In the lower panel, anti-actin was used as a positive loading control. Forty million parasites were used. (**E**) Western blot analysis representing the soluble protein extract from transgenic iKd PfGEXP15 of ring (R), trophozoite (T), and schizont (S) stages. In the lower panel, total protein detected by Ponceau Red staining as loading control. (**F**) Confocal laser scanning microscopy showing GFP-expressing parasites in transfected cultures. Parasite nuclei were stained with DAPI and transgenic parasites express PfGEXP15-GFP-DDD-HA. Merged images showed protein colocalization.

**Figure 7 ijms-24-12647-f007:**
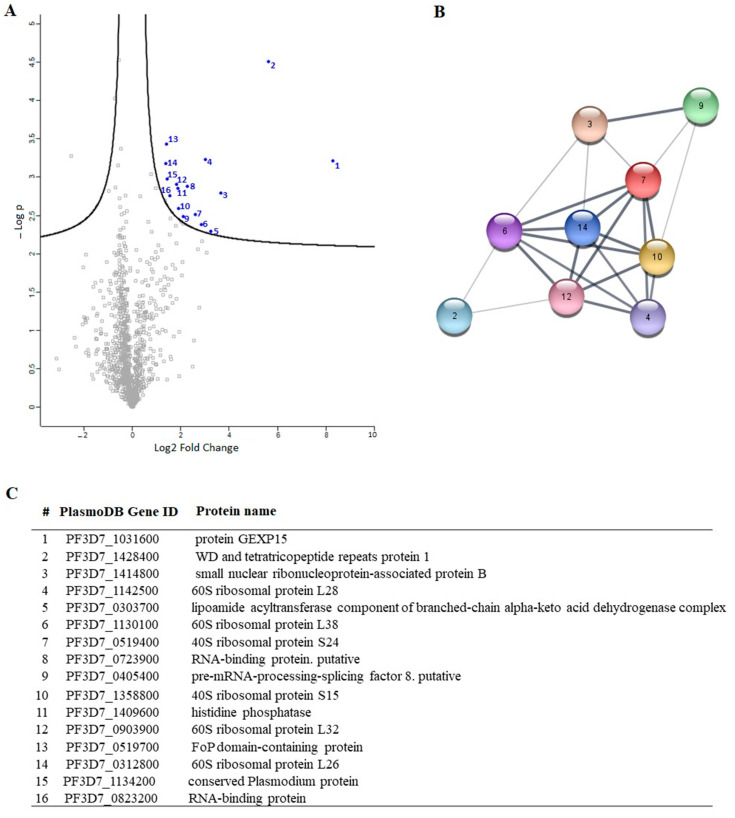
PfGEXP15 interactome analysis. (**A**) Volcano plot representation of PfGEXP15 immunoprecipitation. Blue and gray dots represent statistically significant and non-significant detected proteins respectively. (**B**) STRING network visualization of PfGEXP15-interacting proteins using Cytoscape software (3.9.1). (**C**) List of PfGEXP15 interacting partners. Proteins were ranked according to their Student’s *t*-test difference PfGEXP15–WT in the schizont stage.

**Figure 8 ijms-24-12647-f008:**
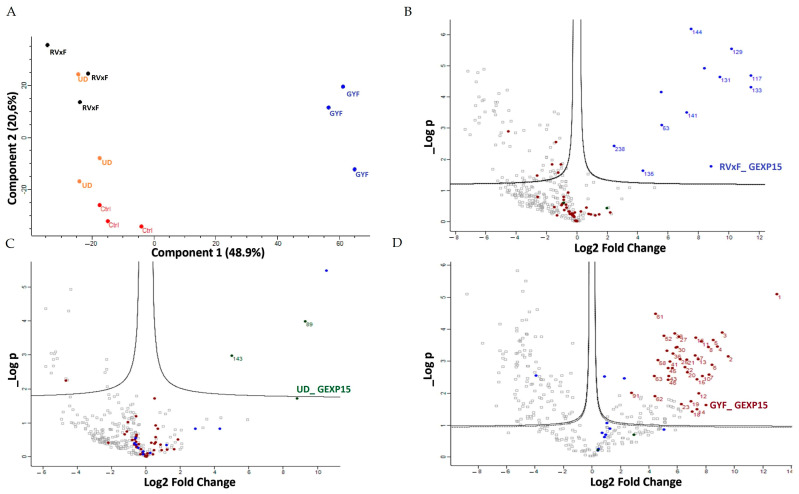
PfGEXP15 pull-down analysis. *(***A**) PCA analysis of the outcome of the pull-downs of PfGEXP15 RVxF (black), UD (orange) and GYF (blue) in three different replicates. The control samples (Ctrl) are indicated in red. Gray dots represent proteins detected with no statistical significance. Volcano plot representation of the outcome of the RVxF (**B**), UD (**C**), and GYF (**D**) pull-downs. The proteins significantly co-purified are indicated in blue, green and purple, respectively.

## Data Availability

The datasets for this study can be found in PRIDE. Project name: Unravelling the function of GEXP15, a regulator of Protein Phosphatase type 1, in *Plasmodium falciparum*. Project accession: PXD042114. Reviewer account details: Username: reviewer_pxd042114@ebi.ac.uk; Password: DonIfsuG.

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
