# Peer review of "Characterization of GEXP15 as a Potential Regulator of Protein Phosphatase 1 in Plasmodium falciparum"

_ijms, 2023, doi:10.3390/ijms241612647_

Round 1

Reviewer 1 Report

The manuscript entitled "Characterization of GEXP15 as a potential regulator of Protein 2 Phosphatase 1 and partner of ribosomal complex in 3 Plasmodium falciparum" reports the discovery of P. falciparum GEXP15 interacts with PP1 and the results of MS for identifying partner proteins of GEXP15. 

However, the same research group already published similar results with P. berghei GEXP15, that amino acid sequence is very similar to Pf GEXP15 (Hollin et al. 2019), so this reviewer could not find significant scientific novelty in this manuscript.

Section 2.1 indicates Pb and Pf GEXP15 are very similar. One cannot find any significant reason the authors should have conducted this research. We know Pf kills humans, but this manuscript needs clear sentences to indicate the importance of this study.

Sections 2.2 and 2.3 are not well connected to the other parts of this manuscript. If the authors want to investigate Dinoflagellata homolog, it is reasonable to show the results of this analysis. It is interesting to compare the homologs RVxF motif +/-. 

Section 2.4

Fig6B Lane1, PfGEXP15 RVxF forms dimer. Is it also the same in the parasite? And it is not so interesting if your input protein is already formed dimer, and both dimer and monomer could bind to GST-PfPP1c (Line 252). The authors cannot indicate the ratio of the dimer and monomer using the figure presented because the signal of the dimer seems saturated.

Fig6B lane 3, Why are free GST bands observed? Please indicate the possible reason.

SDS-PAGE of purified protein stained with CBB should be presented.

Section 2.5 is too preliminary to present. Fig 7 E, since signals of actin in R is saturated, one cannot use this for normalization. The authors should adjust the amount of sample applied to the wells accordingly. 

Section 2.6

They conducted two MS approaches. The first one is not trustable because positive control was not detected. The second one is more relevant. Why should authors consider common proteins detected by the two approaches?

English proof is needed for this manuscript.

Author Response

  • The manuscript entitled "Characterization of GEXP15 as a potential regulator of Protein Phosphatase 1 and partner of ribosomal complex in Plasmodium falciparum" reports the discovery of P. falciparum GEXP15 interacts with PP1 and the results of MS for identifying partner proteins of GEXP15.

However, the same research group already published similar results with P. berghei GEXP15, that amino acid sequence is very similar to Pf GEXP15 (Hollin et al. 2019), so this reviewer could not find significant scientific novelty in this manuscript.

The scientific novelty of this work could be drawn from the data we accumulated throughout this study on GEXP15 in P. falciparum. In this study, we conducted several analyses that were not carried out in Pb, including in silico analysis of the motifs, phylogenetic evolution analysis, and the generation of a 3D model. It is important to highlight that the only common point between PbGEXP15 and PfGEXP15 is their binding to PP1c through the RVxF binding motif, which regulates the phosphatase activity. However, significant functional differences exist between PbGEXP15 and PfGEXP15, primarily related to their localization and interactome. While PbGEXP15 was shown to be present in both nucleus and cytoplasmic compartments, PfGEXP15 is exclusively located in the nucleus. Furthermore, regarding their respective interactomes, PbGEXP15 was found associated with spliceosome and proteosome complexes, as previously reported in Plos Pathogens (Hollin et al., 2019) whereas, PfGEXP15 was found to be associated mainly with ribosomal proteins. This clearly suggested that these proteins have specific functions which can be at least in part explained by their localizations. This underscores the importance of taking precautions when we extrapolate the functions of some proteins and their essentiality from P. berghei to P. falciparum. This is further supported by a review published by Oberstaller et al (2021), which compared the essentiality of genes between P. berghei and P. falciparum. The review revealed that out of 2606 orthologous genes, 93 were found to be mutable in Pf but essential in Pb; while 165 were found to be non-mutable in Pf but dispensable in Pb. This discrepancy underscores the complexity involved in extrapolating protein functions, even among homologs.

  • Section 2.1 indicates Pb and Pf GEXP15 are very similar. One cannot find any significant reason the authors should have conducted this research. We know Pf kills humans, but this manuscript needs clear sentences to indicate the importance of this study.

We thank the reviewer for this comment. Accordingly, we included a new paragraph to indicate why we undertook this study. This paragraph reads:

“Although studies on proteins functions in Pb, the most tractable of the most rodent malaria models for experimental genetics, can provide valuable insights into fundamental aspects of Plasmodium biology, there are limits to how much can be extrapolated to Pf (Oberstaller J 2021 trends in parasitology). For instance, targeted gene-by-gene functional studies showed that the gene encoding Schewanella-like phosphatase (shlp1) in Pf was described as likely essential for erythrocyte development by a functional screen analysis (Zhang M 2018 Science). On the contrary, in Pb, shlp1 is dispensable for the development of blood stage parasites (Patzewitz E M 2013 Cell Rep.).

In Pf, genome-wide saturation mutagenesis suggested GEXP15 as an essential gene in intraerythrocytic developmental cycle. However, the specific roles of this protein throughout the lifecycle of Pf are still not fully characterized.”

  • Sections 2.2 and 2.3 are not well connected to the other parts of this manuscript. If the authors want to investigate Dinoflagellata homolog, it is reasonable to show the results of this analysis. It is interesting to compare the homologs RVxF motif +/-. 

Thank you for your valuable feedback. We appreciate your comments and would like to address your concerns regarding Sections 2.2 and 2.3 of our study, which deal with the 'GEXP15 3D structure modeling' and 'Distribution and phylogenetic analysis of CD2BP2', respectively.

While the primary aim of our study was to perform the molecular and functional characterization of PfGEXP15, we included these sections to provide important contextual information and enhance the overall understanding of our investigation. Moreover, these sections provided a comprehensive characterization of PfGEXP15 within a broader evolutionary context, enriching the significance of our findings. Here's why these sections contribute to the aim of our study:

-GEXP15 3D Structure Modeling (Section 2.2): By employing 3D structure modeling, we aimed to gain insights into the structural features and potential functional domains of PfGEXP15. Understanding the protein's structure can provide crucial clues about its function, interactions, and potential mechanisms of action. This analysis allowed us to explore the putative binding sites, residues involved in protein-protein interactions, and other structural characteristics relevant to PfGEXP15's function.

-Distribution and Phylogenetic Analysis of CD2BP2 (Section 2.3): Investigating the distribution and phylogeny of CD2BP2 homologs and CD2BP2-like protein enabled us to contextualize its evolutionary relationship with related proteins, including homologs in Dinoflagellata. This analysis provided valuable information on the conservation of critical motifs, such as the RVxF motif, which has been implicated in important protein-protein interactions. By comparing the presence or absence of this motif among different homologs, we aimed to elucidate potential functional variations or similarities across species.

The study primarily focused on PfGEXP15, not in Dinoflagellata homologs.

  • Section 2.4

Fig6B Lane1, PfGEXP15 RVxF forms dimer. Is it also the same in the parasite? And it is not so interesting if your input protein is already formed dimer, and both dimer and monomer could bind to GST-PfPP1c (Line 252). The authors cannot indicate the ratio of the dimer and monomer using the figure presented because the signal of the dimer seems saturated.

In this experiment, our main objective was to assess the binding capacity of this motif to the GST-tagged protein. Therefore, whether the RVxF recombinant protein is present as a monomer or dimer doesn’t impact our results. It’s important to note that the formation of dimers in recombinant proteins can occur due to high concentrations of the protein or its improper folding. We do not think that total PfGEXP15 forms dimers as immunoblot from transgenic parasites did not show any band >200 kDa.

Fig6B lane 3, Why are free GST bands observed? Please indicate the possible reason.

The detection of free GST is not unusual in GST pull down experiments. This can be attributed to either intracellular cleavage of the fusion protein or proteolytic cleavage during protein extraction, despite the use of protease inhibitors as well as protease-deficient BL21 DE3 bacterial cells which can result in the binding of this free GST to beads. Furthermore, a translational pausing during protein expression in E. coli can be suspected as well (Tsalkova et al., 1999). However, the use of GST alone (lanes 2, 5 and 8) confirmed that this free GST cannot bind to the different recombinant proteins and the detection of these fragments was only due to GST-PP1.

SDS-PAGE of purified protein stained with CBB should be presented.

The SDS-PAGE of the purified proteins stained with Coomassie blue was added as a Supplementary Figure S4.

  • Section 2.5 is too preliminary to present. Fig 7 E, since signals of actin in R is saturated, one cannot use this for normalization. The authors should adjust the amount of sample applied to the wells accordingly. 

To answer the reviewer’s comment, we removed the actin normalization along with the actin immunoblot and included the loading control obtained by Ponceau red staining of this representative blot. From this immunoblot, it can be seen clearly that the weak detection of PfGEXP15 in rings is not due to a difference in the quantity of total proteins loaded in each lane, suggesting that the protein is accumulated in the later stages and confirming the transcriptomic studies (Oehring, 2012).

  • Section 2.6

They conducted two MS approaches. The first one is not trustable because positive control was not detected. The second one is more relevant. Why should authors consider common proteins detected by the two approaches?

To characterize comprehensively the interactome of PfGEXP15, we conducted two complementary approaches to ensure a more robust characterization of the interactions:

-The first one which is an IP/MS which included soluble extracts from transgenic parasites in order to identify the partners of PfGEXP15 within the parasite. The lack of PfPP1 using the IP of the tagged PfGEXP15 from Pf transgenic parasites can be due to the fact that the complex PfGEXP15-PP1 is unstable at the time point chosen and/or its association is weak/transient. Using this approach, we rather favored the detection of the most abundant and stable complexes.

-The second one is rather a pull-down/MS in which the 3 recombinant proteins containing functional domains were used to pull down partners present in soluble proteins. This can help detecting proteins complexes with transient/weak interaction and/or with low concentrations. Indeed, the second approach which clearly showed the interaction of PfPP1c with the recombinant protein containing the RVXF motif, confirming what we previously reported using Y2H screening and ELISA binding assay (Hollin T, 2016).

Comments on the Quality of English Language

The Manuscript has been now carefully proofread and corrected by a native English speaker.

Reviewer 2 Report

Malaria is an infectious disease that causes recurring episodes of chills and fever. The causative agent is parasites - malaria plasmodia (Plasmodium spp), transmitted by the bites of mosquitoes - carriers of the infection. Plasmodium falciparum causes tropical malaria, the most dangerous variety of the disease; it accounts for more than 90% of all infections. Attacks of fever occur irregularly; in this case, the disease affects the vessels, which can lead first to hypoxia and then to organ failure. This species predominates throughout Africa, and in India, Vietnam, and Thailand is about as common as three-day malaria. Malaria most commonly affects people living or visiting countries with tropical and subtropical climates (Sub-Saharan Africa, the Indian subcontinent, the Solomon Islands, Papua New Guinea, and Haiti). More severe course of malaria affects children under 5 years old, tourists from regions where this disease is not common, visiting subtropical or tropical countries, pregnant women and their unborn children.

The authors conducted a comprehensive investigation into structural and functional characteristics of a conserved Plasmodium-specific regulator called Gametocyte EXported Protein 15, GEXP15 (PF3D7_1031600). 

This study demonstrates the PfGEXP15-PP1 ribosome interaction, which plays a critical role in protein translation. The authors describe in detail the whole process step by step in a high scientific style. If PfGEXP15 could serve as a potential target for malaria drug development, then it could be a major breakthrough in medicine. 

Author Response

Dear reviewer,

Thank you for your positive feedback on our work. We greatly appreciate your valuable comments and insights.

Round 2

Reviewer 1 Report

My comments on the authors are as follows.

---

-GEXP15 3D Structure Modeling (Section 2.2): By employing 3D structure modeling, we aimed to gain insights into the structural features and potential functional domains of PfGEXP15. Understanding the protein's structure can provide crucial clues about its function, interactions, and potential mechanisms of action. This analysis allowed us to explore the putative binding sites, residues involved in protein-protein interactions, and other structural characteristics relevant to PfGEXP15's function.

I'm afraid I have to disagree with the authors' response because this analysis did not connect to the other results in the manuscript. In addition, usually, structural information does not tell us any functional information. This is very true, even in the case of structures derived from crystals or CyroEM. This section with the predicted structures is not important. 

----

-Distribution and Phylogenetic Analysis of CD2BP2 (Section 2.3): Investigating the distribution and phylogeny of CD2BP2 homologs and CD2BP2-like protein enabled us to contextualize its evolutionary relationship with related proteins, including homologs in Dinoflagellata. This analysis provided valuable information on the conservation of critical motifs, such as the RVxF motif, which has been implicated in important protein-protein interactions. By comparing the presence or absence of this motif among different homologs, we aimed to elucidate potential functional variations or similarities across species.

I would like a more specific answer rather than "contextualize". Most of this section is not essential for this manuscript, and they seem to increase the contents with non-essential sections to cover their preliminary experimental results overall.

-----

-The study primarily focused on PfGEXP15, not in Dinoflagellata homologs.

I think so too. But the figure ironically spots the interest of Dinoflagellata homologs rather than Plasmodium spp. Again, what does this figure for?

-----

To answer the reviewer’s comment, we removed the actin normalization along with the actin immunoblot and included the loading control obtained by Ponceau red staining of this representative blot. From this immunoblot, it can be seen clearly that the weak detection of PfGEXP15 in rings is not due to a difference in the quantity of total proteins loaded in each lane, suggesting that the protein is accumulated in the later stages and confirming the transcriptomic studies (Oehring, 2012).

The Ponceau data do not tell us clearly the similar or the same protein quantity. S is almost the same as R. Thus, the protein is not accumulated in the later stages.

-----

-The first one which is an IP/MS which included soluble extracts from transgenic parasites in order to identify the partners of PfGEXP15 within the parasite. The lack of PfPP1 using the IP of the tagged PfGEXP15 from Pf transgenic parasites can be due to the fact that the complex PfGEXP15-PP1 is unstable at the time point chosen and/or its association is weak/transient. Using this approach, we rather favored the detection of the most abundant and stable complexes.

-The second one is rather a pull-down/MS in which the 3 recombinant proteins containing functional domains were used to pull down partners present in soluble proteins. This can help detecting proteins complexes with transient/weak interaction and/or with low concentrations. Indeed, the second approach which clearly showed the interaction of PfPP1c with the recombinant protein containing the RVXF motif, confirming what we previously reported using Y2H screening and ELISA binding assay (Hollin T, 2016).

These answers raise another question. If PfPP1 did not detect after pull-down, how do the authors convince the interaction detected by Y2H or recombinant proteins is true in the parasites?

---

Line 265, I recommend removing "Interestingly"

Line 266, please add "artificial"-dimerized form, and add a sentence to explain it.

Fig. 6., Please add sentences to explain free GST detected in Lane 3, 6, 9 in the main text.

Author Response

GEXP15 3D Structure Modeling (Section 2.2): By employing 3D structure modeling, we aimed to gain insights into the structural features and potential functional domains of PfGEXP15. Understanding the protein's structure can provide crucial clues about its function, interactions, and potential mechanisms of action. This analysis allowed us to explore the putative binding sites, residues involved in protein-protein interactions, and other structural characteristics relevant to PfGEXP15's function.

I'm afraid I have to disagree with the authors' response because this analysis did not connect to the other results in the manuscript. In addition, usually, structural information does not tell us any functional information. This is very true, even in the case of structures derived from crystals or CyroEM. This section with the predicted structures is not important. 

Response:  Thank you for your feedback on our second-round submission. We appreciate your careful consideration of our work. We understand your viewpoint on the importance of structural information for functional inference; and the relevance of the 3D structure modeling section. While we acknowledge that structural information alone may not provide complete functional insights, we firmly believe that it can offer valuable clues and complement other experimental data in understanding protein function, interactions, and mechanisms of action, as presented in the manuscript. We believe that including this analysis enhances the comprehensiveness of our study.

To address your concerns, we would like to highlight that although two proteins with the same overall structural architecture and conserved functional residues can have unrelated functions (J Mol Biol. 2003;331(4):829-60. doi: 10.1016/s0022-2836(03)00734-4.), structural information can still aid in function prediction in several ways (Q Rev Biophys. 2003 Aug;36(3):307-40. doi: 10.1017/s0033583503003901; PLoS Comput Biol. 2008;4(10):e1000160. doi: 10.1371/journal.pcbi.1000160).

For instance:

-Structural similarity between proteins can reveal their common evolutionary origin or functional convergence driven by common constraints, even in the absence of significant sequence similarity. Lipocalins (Sci Rep. 2016;6:32372. doi: 10.1038/srep32372.) and viral RNA dependent polymerases (PLoS One. 2014;9(5):e96070. doi: 10.1371/journal.pone.0096070.) provide examples of structural and functional convergence.

-Structural motifs can indicate binding sites. Residues forming functional signatures, although not necessarily adjacent in sequence, tend to cluster in the 3D structure, forming binding sites for various molecules (Nature. 2022;604(7907):763-770. doi: 10.1038/s41586-022-04619-y.). For example, the structural helix-turn-helix motif is well known to be found in DNA binding proteins that regulate gene expression (FEMS Microbiol Rev. 2005 Apr;29(2):231-62. doi: 10.1016/j.femsre.2004.12.008.).

-Residues with similar functions in different proteins often possess similar physicochemical characteristics. For example, residues involved in DNA binding share common structural and physicochemical features in DNA-binding proteins (e.g., secondary structures, geometries, solvent accessibility, charge, hydrophobicity). Characterizing and quantifying these features can aid in predicting functional residues (Proc Natl Acad Sci USA. 2015;112(52):15910-5. doi: 10.1073/pnas.1518946112; Proteins. 2016;84(8):1147-61. doi: 10.1002/prot.25061).

-Subcellular localization knowledge narrows down the potential functions of a protein and is relevant for experimental characterization. Predicting subcellular localization can be achieved through homology and motif analysis (BMC Cell Biol. 2010;11:74. doi: 10.1186/1471-2121-11-74).

Not only structure led to functionally relevant information, but also structural analysis can reveal insights that cannot be experimentally determined. For example, all-atom molecular dynamics simulations (derived from 3D models) revealed that one amino acid mutation repels a water molecule that coordinates the position of a metal ion cofactor (J Virol. 2017;91(21):e01028-17. doi: 10.1128/JVI.01028-17).

In order to keep our data accessible to the readers of IJMS, we removed the Figure 3 from the main text of the Ms and included it as a supplementary figure (Supplementary Figure S3).

----

-Distribution and Phylogenetic Analysis of CD2BP2 (Section 2.3): Investigating the distribution and phylogeny of CD2BP2 homologs and CD2BP2-like protein enabled us to contextualize its evolutionary relationship with related proteins, including homologs in Dinoflagellata. This analysis provided valuable information on the conservation of critical motifs, such as the RVxF motif, which has been implicated in important protein-protein interactions. By comparing the presence or absence of this motif among different homologs, we aimed to elucidate potential functional variations or similarities across species.

I would like a more specific answer rather than "contextualize". Most of this section is not essential for this manuscript, and they seem to increase the contents with non-essential sections to cover their preliminary experimental results overall.

Response: The section "Distribution and phylogenetic analysis of CD2BP2" provides essential insights into the evolutionary relationships, phylogenetic distribution, and conservation of critical motifs, such as the RVxF motif, within CD2BP2 homologs and CD2BP2-like proteins across diverse Metazoan species.

Moreover, the phylogenetic analysis presented in this section sheds light on the evolutionary divergence and convergent evolution of CD2BP2 and CD2BP2-like proteins. The clustering of these proteins into distinct groups provides insights into their evolutionary history and highlights the conserved structural features that may be indicative of functional similarities.

We regret that the reviewer does not fully appreciate the significance of these insights. However, it is important to note that the relevance of this analysis extends beyond the immediate perspective. Understanding the evolutionary context and conservation of CD2BP2 and its homologs enhances our comprehension of their functional implications in various species. These findings contribute to the broader scientific understanding of CD2BP2 and its evolutionary significance, irrespective of their direct implications for the specific focus of the manuscript.

-----

-The study primarily focused on PfGEXP15, not in Dinoflagellata homologs.

I think so too. But the figure ironically spots the interest of Dinoflagellata homologs rather than Plasmodium spp. Again, what does this figure for?

Response: The inclusion of Dinoflagellata in the figure was meant to highlight the broader distribution of the RVxF motif, unknown domain, GYF domain, as well as the presence of CD2BP2 and CD2BP2-like proteins across various (i.e., 17) eukaryotic lineages, rather than to simply emphasize their relevance specifically to Plasmodium spp.  

Based on the figure, it is evident that CD2BP2 homologs in Metazoa display the presence of the RVxF motif, which is known to be indicative of PP1 binding activity. Conversely, none of the CD2BP2-like proteins in the depicted eukaryotic lineages exhibit the RVxF motif, except for Apicomplexa and Oomycota.

Dinoflagellata as a sister taxon of Apicomplexa (DOI: 10.1073/pnas.1423790112), lack as well the ‘RVxF motif’ which makes its presence in Apicomplexa even more noteworthy.

This particular observation holds significant importance in comprehending the evolutionary convergence of CD2BP2-like proteins specifically within the Apicomplexa lineage, this is to say within Plasmodium. Understanding the evolutionary convergence of CD2BP2-like proteins is crucial for unraveling the mechanisms underlying functional adaptations across diverse eukaryotic lineages. The presence of the RVxF motif in Apicomplexa, despite its absence in most other lineages, suggests a unique evolutionary trajectory for CD2BP2-like proteins within the Apicomplexa group. Investigating the functional implications and molecular interactions associated with this convergence can provide valuable insights into the specific adaptations and regulatory mechanisms that have shaped the biology of Apicomplexa organisms.

To address the reviewer’s concern regarding the insistence on Dinoflagellata, maybe this impression was due to the frequency of mentioning it in the text, we will replace this sentence:

“However, CD2BP2-like proteins in dictyostelids, chlorophytes, and dinoflagellates lacked the RVxF motif,”

By
“However, CD2BP2-like proteins in Rhizaria, Plantae and Amoebozoa lacked the RVxF motif”

-----

To answer the reviewer’s comment, we removed the actin normalization along with the actin immunoblot and included the loading control obtained by Ponceau red staining of this representative blot. From this immunoblot, it can be seen clearly that the weak detection of PfGEXP15 in rings is not due to a difference in the quantity of total proteins loaded in each lane, suggesting that the protein is accumulated in the later stages and confirming the transcriptomic studies (Oehring, 2012).

The Ponceau data do not tell us clearly the similar or the same protein quantity. S is almost the same as R. Thus, the protein is not accumulated in the later stages.

Response: To better clarify this point, we replaced the term “later stages” with “trophozoites” for more precise description.

-----

-The first one which is an IP/MS which included soluble extracts from transgenic parasites in order to identify the partners of PfGEXP15 within the parasite. The lack of PfPP1 using the IP of the tagged PfGEXP15 from Pf transgenic parasites can be due to the fact that the complex PfGEXP15-PP1 is unstable at the time point chosen and/or its association is weak/transient. Using this approach, we rather favored the detection of the most abundant and stable complexes.

-The second one is rather a pull-down/MS in which the 3 recombinant proteins containing functional domains were used to pull down partners present in soluble proteins. This can help detecting proteins complexes with transient/weak interaction and/or with low concentrations. Indeed, the second approach which clearly showed the interaction of PfPP1c with the recombinant protein containing the RVXF motif, confirming what we previously reported using Y2H screening and ELISA binding assay (Hollin T, 2016).

These answers raise another question. If PfPP1 did not detect after pull-down, how do the authors convince the interaction detected by Y2H or recombinant proteins is true in the parasites?

Response:

We understand that the lack of detection of PfPP1 in the PfGEXP15 IP/MS may raise the question about the interaction between both proteins in the parasite, as we do not have direct experimental evidence in this study. However, we demonstrated the likelihood of this interaction via the RVxF motif by Y2H, GST pull-down, and pull-down experiments in particular with the recombinant RVxF-contianing protein in this study, in correlation with findings from our previous studies (Hollin et al., 2016), Supporting this is the fact that in P. berghei parasites, PbGEXP15 was also detected among the top PP1-interacting proteins in both schizont and gametocyte stages (De Witte et al., 2022). And the reciprocal IP/MS identified PbPP1 after PbGEXP15 immunoprecipitation (Hollin et al., 2019). Altogether, these results demonstrate without any doubt the binding between the two proteins.

---

Line 265, I recommend removing "Interestingly"

The sentence has been corrected.

Line 266, please add "artificial"-dimerized form, and add a sentence to explain it.

This sentence has been added to explain the dimerization:

An artificial-dimerized form of PfGEXP15, able to bind to PfPP1c (Figure 5B), was detected and could be due to the overexpression or misfolding of the recombinant fragment produced.

Fig. 6., Please add sentences to explain free GST detected in Lane 3, 6, 9 in the main text.

This sentence has been added in the figure legend:

The detection of free GST in lanes 3, 6 and 9 could be attributed to non-specific cleavage or protease activity.

Round 3

Reviewer 1 Report

We understand that the lack of detection of PfPP1 in the PfGEXP15 IP/MS may raise the question about the interaction between both proteins in the parasite, as we do not have direct experimental evidence in this study. However, we demonstrated the likelihood of this interaction via the RVxF motif by Y2H, GST pull-down, and pull-down experiments in particular with the recombinant RVxF-contianing protein in this study, in correlation with findings from our previous studies (Hollin et al., 2016), Supporting this is the fact that in P. berghei parasites, PbGEXP15 was also detected among the top PP1-interacting proteins in both schizont and gametocyte stages (De Witte et al., 2022). And the reciprocal IP/MS identified PbPP1 after PbGEXP15 immunoprecipitation (Hollin et al., 2019). Altogether, these results demonstrate without any doubt the binding between the two proteins.

This is very doubtful since the authors explained that Pb and Pf are different in earlier responses. Anyway, they need to add these sentences in the discussions.

According to their explanation of MS, I think the results are insufficient to say that they characterized it as a partner of the ribosomal complex. I recommend removing "and partner of ribosomal complex" from the title.

Author Response

The difference observed between Pb and Pf seems to concern the partners associated with the GYF interaction domain. The binding between GEXP15 and PP1 is mediated via the RVxF motif, well described as the major PP1 binding site for regulatory subunits. This motif is well conserved in Plasmodium spp. and all complementary approaches perforned in vitro confirmed a similar binding between the two proteins. We don't think that there is a difference in Pf and Pb regarding this aspect.

We have included the previous explanation in the discussion.

As suggested by the reviewer, we removed "and partner of ribosomal complex" from the title.